# Gender and Age Influence in Pre-Competitive and Post-Competitive Anxiety in Young Tennis Players

**Rafael Martínez-Gallego** [1], **Santos Villafaina** [2,3], **Miguel Crespo** [4,*] and **Juan Pedro Fuentes-García** [2]

1   Department of Sport and Physical Education, University of Valencia, 46010 Valencia, Spain;
    rafael.martinez-gallego@uv.es
2   Faculty of Sports Science, University of Extremadura, Avda: Universidad S/N, 10003 Cáceres, Spain;
    svillafaina@unex.es (S.V.); jpfuent@unex.es (J.P.F.-G.)
3   Department of Sport and Health, School of Health and Human Development, University of Evora,
    7004-516 Evora, Portugal
4   Development Department, International Tennis Federation, London SW15 5XZ, UK
*   Correspondence: miguel.crespo@itftennis.com

**Abstract:** To study the influence of age and gender on pre-competitive and post-competitive anxiety and self-confidence in young tennis players. A total of 42 U'12 to U'18 category tennis players, 12 females and 30 males, participated in this cross-sectional study. The players had a mean age of 13.74 (2.07) years old and a national competitive experience of 4.00 (2.14) years. The pre-competitive anxiety of the participants was assessed using the Competitive State Anxiety Inventory–2R (CSAI-2R) and the State Trait Anxiety Inventory (STAI-E) before and after an official tournament's match. Results showed that younger players showed lower trait anxiety (r = 0.333; $p < 0.05$), lower pre-match state anxiety (r = 0.501; $p < 0.01$) and lower pre-match somatic anxiety (r = 0.313; $p < 0.05$). Furthermore, girls exhibited higher values of state anxiety (r = 0.445; $p < 0.01$) and somatic anxiety (r = 0.440; $p < 0.01$) than boys before the match. However, differences were not observed in the trait anxiety measured by STAI-E (r = 0.203; $p = 0.213$), cognitive anxiety (r = 0.140; $p = 0.363$), and self-confidence measured by the CSAI questionnaire (r = 0.150; $p = 0.333$), before the match. Therefore, coaches and sport psychologists should implement adequate on- and off-court individualized interventions to manage anxiety, specifically in girls and players over 14 years old. Although anxiety levels were similar to those before the COVID-19 pandemic, due to the influence of the pandemic on mental health, results might be taken with caution.

**Keywords:** racket sports; juniors; stress; pressure; choking

## 1. Introduction

The psychological demands of tennis have been extensively identified [1] and related to its different aspects at both participation and performance levels [2]. Anxiety has been identified as one the main mental challenges of the game at all levels of play. In this regard, the ability of the players to deal with pressure situations is directly related to their performance [3,4]. Furthermore, the probability of an error in performance increases significantly with the pressure of the point [5]. Thus, previous studies have explored anxiety in tennis among different age groups, playing standards, as well as gender differences, using a variety of assessment tools [6–8]. In addition, COVID-19 pandemic considerable consequences for tennis players and other stakeholders [9–11], increasing anxiety [12,13].

Previous studies have found that age of tennis players can be related to anxiety. In this regard, Ebbeck [7] showed that age negatively related to competitive trait anxiety in junior tennis players. However, parents and coaches can influence anxiety in athletes [14,15] since they are usually focused on "win at all cost" [14]. In this line, Salman & Mahmoud [16] found negative correlations between cognitive anxiety and the motivation to achieve success, and positive correlation with the motivation to avoid failure. This could be the

reason why a previous study found that 12–16-year-old tennis players showed moderate scores of anxiety Santos-Rosa et al. [17]. Interestingly, Koehn [18] confidence protected players against their interpretations of anxiety.

Previous studies have studied the differences in anxiety between male and female tennis player during and precompetition. Regarding precompetitive anxiety, previous studies have reported that female athletes reported higher levels of anxiety than their male counterparts due to an increase of somatic symptoms and a decline in self-confidence before competition [19]. Similarly, Khot & Bujurke [20] found that female players had higher anxiety and stress levels than male. Cohen-Zada et al. [21] assessed the performance of female and male professional tennis players when competing in matches under pressure with high monetary rewards. During the match, men were consistently found choking under competitive pressure. However, women showed a reduction in performance in key moments, it was never 50% smaller than in the case of men. De Paola & Scoppa [22] showed that women who lost the first set would play poorly the second set much more likely than men did. Women also showed more disappointment when under pressure of receiving negative feedback and being behind. However, there are studies which did not find differences between male and female tennis players. In this line, Keskin et al. [23] did not find significant differences between adults male and female tennis players in the state anxiety (somatic and cognitive). Similarly, a previous study found that females showed less competitive anxiety than males with increasing age [7].

As shown above, anxiety in tennis has been an extensively studied topic in tennis due to its impact on the performance. However, specific studies with young tennis players which focus on gender and age differences are scarce. Therefore, this study aimed to analyse the pre- and post-competitive anxiety of young tennis players and its gender and age implications.

Based on the literature review on the effects of competition on athletes' anxiety, it was hypothesised that:

1.  Young tennis players would have higher anxiety and lower self-confidence values before than after the competition.
2.  Pre-competitive anxiety values of young tennis players would correlate positively, as well as post-competitive anxiety values. However, they would correlate negatively with self-confidence.
3.  Younger tennis players would have lower values of anxiety and higher values of self-confidence than older players.
4.  Boys tennis players would have lower anxiety values and higher self-confidence values than the girl counterparts.

## 2. Materials and Methods

### 2.1. Sample

A total of 42 U'12 to U'18 category tennis players, 12 females and 30 males, participated in this cross-sectional study. Table 1 shows the main characteristics of the players who participated in the study. Participants had a mean age of 13.74 (2.07) years and a competitive experience of 4.00 (2.14) years in national competitions.

**Table 1.** Characteristics of the tennis players.

| Variable | Mean (SD) |
| --- | --- |
| Age (years) | 13.74 (2.07) |
| Height (cm) | 162.45 (11.66) |
| Weight (kg) | 50.79 (13.56) |
| Body Mass Index (kg/m$^2$) | 18.9 (2.79) |
| National competition experience (years) | 4.00 (2.14) |
| Number of training days/week | 3.55 (0.92) |
| Number of training hours/day | 1.26 (0.40) |

### 2.2. Procedure

Participants were evaluated before, during and after a tennis match played (in an official tournament valid for the national ranking).

The levels of anxiety and self-confidence were evaluated before and immediately after the tennis match. It was also checked that no product or substance was ingested by tennis players in the sample which could influence their nervous system 24-h before performing the protocol.

The Committee of Ethics of the University approved the study procedures (approval number: 112/2021). The appropriate written consent to participate in this study was also signed by the players.

### 2.3. Instruments

The Spanish version of the Competitive State Anxiety Inventory–2R (CSAI-2R) was employed to measure the participants' pre-competitive anxiety level [24,25]. This questionnaire has been shown extremely useful in sports [26,27], and even military [28] contexts. The levels of cognitive and somatic anxiety and the self-confidence were calculated from the 17 items of the questionnaire. Each item was assessed by a 4-point Likert scale ranging from "not at all" to "very much so". The negative feelings about the performance and the consequences of the performance were assessed with 5 items, and the overall score ranging from 5 to 20 points, of the Cognitive Anxiety subscale. The 7 items of the Somatic Anxiety subscale, with the minimum score being 7 and the maximum 28, measured the increased heart rate, match discomfort, sweating and muscle tension, which are physiological indicators of the anxiety's perception. The 5 items of the self-confidence subscale, with an overall score between 5 and 20 was used to measure the degree of confidence on the athletes´ competitive success.

The Spanish version of the State Trait Anxiety Inventory (STAI-E) [29], a reliable tool to study variations in anxiety which are rapid state-dependent [30], was used to measure anxiety. This questionnaire consists of two scales: A-Trait (A-T) and A-State (A-S), with 20 items each. A Likert scale from 0 (almost never) to 3 (almost always) is used to rate the 40 items of the questionnaire. The first scale, A-T, describes how the participant feels at a "particular moment". It indicates a relatively stable anxious propensity, and when individuals perceive situations as threatening an increase in their A-S is observed. The second scale (A-S) describes the transient emotional state of the individual, characterised by perceived feelings of apprehension and tension, which are subjective, and conscious. It also shows the autonomic nervous system's hyperactivity, which can fluctuate in intensity and vary over time.

So, the negative scale was subtracted from the positive scale and 30 was added to the result. It should be noted that, in the Spanish version of the STAI used in this study, the response scale was changed (the original 0–4 was reduced to 0–3), which reduced by 20 the averages (means). In this regard, values of the Spanish version plus 20 points were included. This change did not affect the other statistics (standard deviation, reliability, and correlation indices, etc.) so that they could be compared directly. The score range for the test was 20–80, with a higher score representing a higher level of anxiety.

### 2.4. Statistical Analysis

The SPSS statistical package (Statistical Package for Social Sciences, version 25 for Windows, IBM Corporation, Armonk, NY, USA) was used to conduct the statistical analysis. After the Shapiro-Wilk test results, non-parametric tests were conducted.

The internal consistency of the questionnaires was conducted using the Cronbach's alpha reliability analysis, with values greater than or equal to 0.70 [31]. Furthermore, for greater accuracy the omega coefficient [32], was used to check the internal consistency of the variables in the study [33]. The highest values of the McDonald omega coefficient, with a range between 0 and 1, provide the most reliable measurements [33]. When using the

omega coefficient, Campo-Arias and Oviedo [34] suggest that a value of confidence greater than 0.70 is acceptable.

The differences between high and low-demanding matches in the cognitive, somatic and self-confidence subscales of the CSAI-2R questionnaires were examined using non-parametric statistics (Wilcoxon signed rank test).

The Wilcoxon signed-rank test was used to test the difference between the pre- and post-measures for each variable. Effect sizes [r] for the non-parametric tests were: 0.1 was a small effect, 0.3 was medium, and 0.5 was large [35,36].

Spearman's correlation thresholds were used to conduct a bivariate correlation analysis between all the psychological profile and perceived stress variables.

### 3. Results

Table 2 shows the mean of cognitive anxiety, somatic anxiety self-confidence, state anxiety and trait anxiety pre- and post-match. Results showed higher levels of cognitive anxiety, somatic anxiety and state anxiety before the match. Regarding internal consistency of the questionnaires, the results of our study suggest an adequate level of internal consistency, with Cronbach's alpha and McDonald omega coefficient ranges in all variables, with the exception of "State anxiety (Negative scale)" values greater than or equal to 0.70. It should be noted that a significant number of variables present high values of internal consistency, close to or even, in some cases, higher than 0.90.

**Table 2.** Descriptive statistics and reliability analysis.

| | Variables | M | SD | α | ω |
|---|---|---|---|---|---|
| Pre | Cognitive anxiety | 2.31 | 0.86 | 0.88 | 0.88 |
| Post | | 2.21 | 0.87 | 0.87 | 0.88 |
| Pre | Somatic anxiety | 1.71 | 0.56 | 0.82 | 0.84 |
| Post | | 1.67 | 0.72 | 0.89 | 0.91 |
| Pre | Self-confidence | 2.90 | 0.57 | 0.76 | 0.76 |
| Post | | 2.90 | 0.67 | 0.84 | 0.83 |
| Pre | State anxiety (Positive scale) | 0.72 | 0.48 | 0.88 | 0.88 |
| Post | | 0.66 | 0.71 | 0.93 | 0.93 |
| Pre | State anxiety (Negative scale) | 1.96 | 0.54 | 0.88 | 0.88 |
| Post | | 2.06 | 0.66 | 0.89 | 0.90 |
| Pre | State anxiety (Total) | 37.62 | 9.83 | - | - |
| Post | | 36 | 13.1 | - | - |
| | Trait anxiety (Positive scale) | 0.85 | 0.85 | 0.76 | 0.79 |
| | Trait anxiety (Negative scale) | 2.16 | 0.38 | 0.60 | 0.60 |
| | Trait anxiety (Total) | 30.92 | 7.05 | - | - |

M: Mean, SD: standard deviation, α: Cronbach's alpha, ω: omega coefficient.

Table 3 shows the summary statistics and inferential analysis for anxiety and self-confidence levels before and after the competitive matches. Although before the competition matches the values of state, cognitive and somatic anxiety were slightly higher, pre- and post-match measures did not show significant differences. Self-confidence stayed at the same level before and after the match.

Figure 1 shows the correlations between the different anxiety and self-confidence variables before and after the matches. State anxiety correlated positively with the other anxiety-related variables, and negatively with the variables related to self-confidence. The pre-match values of the anxiety variables correlated positively, while they correlated negatively with self-confidence. Similarly, the values found for the anxiety-related variables after the matches correlated positively with each other, and negatively with self-confidence. Furthermore, pre-and post-match cognitive anxiety and pre- and post-match self-confidence also showed correlations.

**Table 3.** Acute effects of competition on anxiety and self-confidence.

| Variables | Pre Mean (SD) | Post Mean (SD) | *p*-Value | Effect Size |
|---|---|---|---|---|
| STAI-E A-S | 37.62 (9.83) | 36.00 (13.08) | 0.452 | 0.127 |
| Cognitive anxiety | 2.31 (0.86) | 2.21 (0.87) | 0.435 | 0.132 |
| Somatic anxiety | 1.71 (0.56) | 1.67 (0.72) | 0.896 | 0.022 |
| Self-confidence | 2.90 (0.57) | 2.90 (0.67) | 0.991 | 0.001 |

STAI-E A-S: State Trait Anxiety Inventory A-State; SD: Standard Deviation.

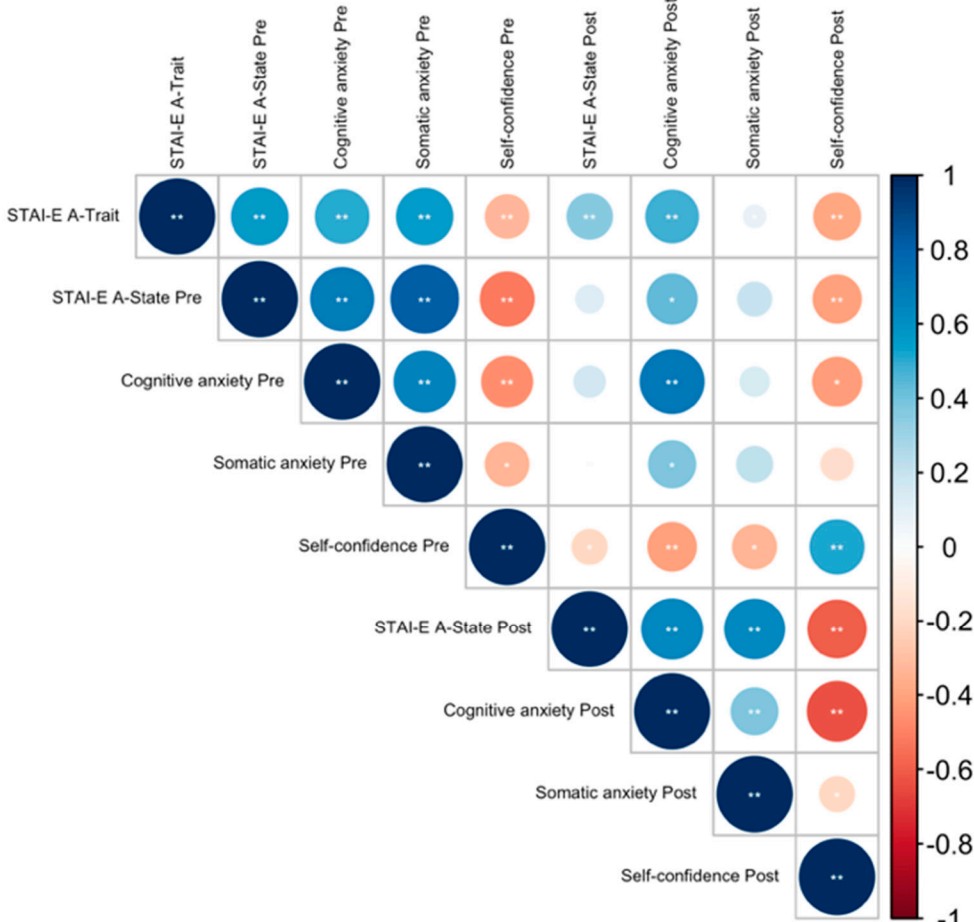

**Figure 1.** Correlations between pre and post-competitive values of the variables related with anxiety and self-confidence. (* *p* < 0.05; ** *p* < 0.01).

Table 4 shows the inferential analysis between the group of players aged 14 years old and under, and the group of players older than 14 years old. Compared to the players older than 14 years, the younger players showed lower trait anxiety (r = 0.333; *p* < 0.05), lower pre-match state anxiety (r = 0.501; *p* < 0.01) and lower pre-match somatic anxiety (r = 0.313; *p* < 0.05) levels. In addition, there is also a clear trend indicating that younger players showed higher self-confidence after the matches (r = 0.330; *p* = 0.05) than older ones.

Table 5 shows the comparisons regarding gender. Girls exhibited higher values of state anxiety (r = 0.445; *p* < 0.01) and somatic anxiety (r = 0.440; *p* < 0.01) than boys before the match. However, differences were not observed in the trait anxiety measured by STAI-E (r = 0.203; *p* = 0.213), and the cognitive anxiety (r= 0.140; *p* = 0.363), and self-confidence measured by the CSAI questionnaire (r = 0.150; *p* = 0.333), before the match. Regarding post-match anxiety and self-confidence variables, statistically significant differences (*p* > 0.05) were not observed between girls and boys.

**Table 4.** Age-related differences in young tennis players' anxiety and self-confidence.

|  | Variables | 14 Years Old and Under Mean (SD) | Over 14 Years Old Mean (SD) | *p*-Value | Effect Size |
|---|---|---|---|---|---|
|  | STAI–E A-T | 29.74 (7.46) | 33.40 (3.57) | 0.040 * | 0.333 |
| Pre | STAI–E A-S | 34.48 (6.28) | 44.00 (9.18) | 0.001 * | 0.501 |
| Post |  | 36.09 (13.97) | 36.80 (12.55) | 0.521 | 0.109 |
| Pre | Cognitive anxiety | 2.21 (0.80) | 2.64 (0.98) | 0.443 | 0.242 |
| Post |  | 2.16 (0.93) | 2.44 (0.75) | 0.301 | 0.175 |
| Pre | Somatic anxiety | 1.58 (0.47) | 1.97 (0.64) | 0.043 * | 0.313 |
| Post |  | 1.68 (0.74) | 1.66 (0.78) | 0.801 | 0.043 |
| Pre | Self-confidence | 2.91 (0.60) | 2.88 (0.50) | 0.443 | 0.119 |
| Post |  | 3.01 (0.69) | 2.60 (0.60) | 0.051 | 0.330 |

SD: Standard Deviation; STAI-E A-T: State Trait Anxiety Inventory A-Trait; STAI-E A-S: State Trait Anxiety Inventory A-State; * *p*-value < 0.05.

**Table 5.** Gender-related differences in young tennis players' anxiety and self-confidence.

|  | Variables | Girls Mean (SD) | Boys Mean (SD) | *p*-Value | Effect Size |
|---|---|---|---|---|---|
|  | STAI–E A-T | 31.56 (6.25) | 30.58 (6.96) | 0.212 | 0.203 |
| Pre | STAI–E A-S | 43.22 (8.36) | 35.17 (7.44) | 0.004 * | 0.445 |
| Post |  | 35.78 (16.46) | 36.50 (12.41) | 0.143 | 0.248 |
| Pre | Cognitive anxiety | 2.51 (1.14) | 2.28 (0.76) | 0.363 | 0.140 |
| Post |  | 2.16 (0.96) | 2.28 (0.86) | 0.410 | 0.139 |
| Pre | Somatic anxiety | 2.02 (0.60) | 1.58 (0.48) | 0.004 * | 0.440 |
| Post |  | 1.56 (0.80) | 1.72 (0.73) | 0.517 | 0.110 |
| Pre | Self-confidence | 2.87 (0.62) | 2.92 (0.56) | 0.333 | 0.150 |
| Post |  | 2.96 (0.66) | 2.86 (0.70) | 0.532 | 0.106 |

SD: Standard Deviation; STAI-E A-T: State Trait Anxiety Inventory A-Trait; STAI-E A-S: State Trait Anxiety Inventory A-State; * *p*-value < 0.05.

## 4. Discussion

The purpose of this research was to study the influence of age and gender in young tennis players' anxiety and self-confidence pre- and post-competition. The results showed higher levels of pre-competitive anxiety in older players with respect to younger players. Similarly, higher values of pre-competitive anxiety were also found in girls as compared to boys.

The first hypothesis was supported partially. Although the results obtained showed higher anxiety levels in all dimensions and lower self-confidence levels before than after the matches, there were no significant differences in any of the variables. Only one study has previously analysed pre- and post-competitive anxiety and self-confidence in female elite tennis players [37]. Authors reported significant differences, except for somatic anxiety in the winners of the matches. Previous researches have studied team sports such as basketball [38], football [39,40] or volleyball [41], and older athletes than those in the present study. Several studies have shown differences in anxiety levels between individual and team sports [19,42] and according to age [43,44], so this may be the reasons why the results of our study are different to the previous ones. In addition, data was assessed post COVID-19 outbreak. Since COVID-19 pandemic have been a significant impact on anxiety, results might be influenced by this issue. Nevertheless, post-competitive levels of anxiety are similar to those obtained prior to COVID-19 pandemic [18].

The second hypothesis which stated that all anxiety values would correlate positively among themselves, and would correlate negatively with self-confidence, was supported by the results. Pre-competitive state anxiety, cognitive anxiety and somatic anxiety all corre-

lated positively with each other, while they all correlated negatively with pre-competitive self-confidence. Likewise, post-competitive state anxiety, cognitive anxiety and somatic anxiety correlated positively with each other, while all of them correlated negatively with post-competitive self-confidence. These results are in line with those of Fernández et al. [37], in which relationships between anxiety and self-confidence values in female tennis players before and after the matches were found. Moreover, they are also similar to the results of previous research with athletes in other individual sports such as gymnastics [45].

As related to the third hypothesis, in which it was stated that 14 years old and under players would have lower anxiety and higher self-confidence than over 14 years old players, the results obtained partially support this hypothesis. Players younger than 14 years old showed lower state and somatic anxiety levels before matches than their older counterparts. In addition, younger players showed a higher self-confidence after the matches than older players. Previous studies that analysed racket sports such as paddle tennis, with players of similar ages [43], found that younger players were those who showed greater self-confidence, while their older counterparts showed higher cognitive and somatic anxiety. Therefore, the results found in our study support those of Cayetano et al. (2017), except for somatic anxiety levels in which no differences were found as a function of age. This anxiety increase and the self-confidence decrease with age could be explained by the greater desire to win that adolescent players showed as compared to younger children [46].

As per the fourth hypothesis, which referred to differences related to the players' gender, the results found partially support what was hypothesised. Although no gender differences were found in self-confidence and cognitive anxiety levels, it was found that girls had higher levels of state anxiety and somatic anxiety than boys. These findings are similar to those from Filaire et al. [47] with adult tennis players, in which they reported higher somatic anxiety in female players compared to male players, and found no gender differences in self-confidence levels. However, Keskin et al. [23] found no differences between genders in either somatic or cognitive anxiety of adult tennis players.

Although this study has followed the methodology used in many previous studies in the same area, certain limitations can be identified. Firstly, anxiety was assessed using questionnaires, and no other psychophysiological measures that could complement the results obtained were used. Therefore, future studies could employ other tools to assess anxiety levels in junior tennis players, which could complement our results. On the other hand, although the main goal of the research was to test the gender and age influence on the pre- and post-competition anxiety of tennis players, the outcome of the matches was not considered. Future studies should investigate the optimal level of anxiety or activation that does not negatively influence performance. Therefore, future studies could consider including this variable to examine its influence on levels of anxiety and self-confidence of junior tennis players. Research on post COVID-19 recovery strategies could also be of interest to better understand how the mental health and the performance of these players have been impacted by the pandemic.

The results of this research have practical applications. Coaches and sport psychologists should understand the multi-faceted characteristics of anxiety in young tennis players and implement the adequate on- and off-court interventions according to the specific needs of their players. Specifically, as it relates to the differences between genders and age groups, our results provide direction towards the implementation of individualized approaches which are recommended when dealing with junior players of different genders and age groups.

## 5. Conclusions

Players under 14 years old showed lower values of pre-competitive state anxiety and pre-competitive somatic anxiety and higher post-competitive self-confidence than players over 14 years old. Regarding gender, girls reported higher values of pre-competitive state anxiety and pre-competitive somatic anxiety than boys. Therefore, coaches and sport psychologists should implement adequate on- and off-court individualized interventions

to manage anxiety, specifically in girls and players over 14 years old. Although anxiety levels were similar to those before the COVID-19 pandemic, due to the influence of the pandemic on mental health, results might be taken with caution.

**Author Contributions:** Conceptualization, R.M.-G., J.P.F.-G., M.C. and S.V.; methodology, R.M.-G. and J.P.F.-G.; formal analysis, R.M.-G., J.P.F.-G., M.C. and S.V.; investigation, R.M.-G. and M.C.; resources M.C.; data curation, R.M.-G. and J.P.F.-G.; writing—original draft preparation, R.M.-G., J.P.F.-G., M.C. and S.V.; writing—review and editing, R.M.-G., J.P.F.-G., M.C. and S.V.; supervision, J.P.F.-G. and S.V.; project administration, J.P.F.-G.; funding acquisition, J.P.F.-G. All authors have read and agreed to the published version of the manuscript.

**Funding:** The author JPFG was supported by a grant from the Department of Economy and Infrastructure of the Junta de Extremadura through the European Regional Development Fund. A way to make Europe (GR21094). The author SV was supported by a grant from the Universities Ministry of Spain and the European Union (NextGenerationUE) "Ayuda del Programa de Recualificación del Sistema Universitario Español, Modalidad de ayudas Margarita Salas para la formación de jóvenes doctores" (MS-03).

**Institutional Review Board Statement:** The study was conducted in accordance with the Declaration of Helsinki, and approved by the Ethics Committee of University of Extremadura (approval number: 112/2021).

**Informed Consent Statement:** Informed consent was obtained from all subjects involved in the study.

**Data Availability Statement:** The data presented in this study are available on request from the corresponding author. The data are not publicly available due to privacy.

**Acknowledgments:** This study has been made thanks to the contribution of the International Tennis Federation as well as the Department of Economy and Infrastructure of the Junta de Extremadura through the European Regional Development Fund. A way to make Europe (GR21094). The author SV was supported by a grant from the Universities Ministry of Spain and the European Union (NextGenerationUE) "Ayuda del Programa de Recualificación del Sistema Universitario Español, Modalidad de ayudas Margarita Salas para la formación de jóvenes doctores" (MS-03).

**Conflicts of Interest:** The authors certify that there is no conflict of interest with any financial organization regarding the material discussed in the manuscript.

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
