# Peer review of "Gender and Age Influence in Pre-Competitive and Post-Competitive Anxiety in Young Tennis Players"

_sustainability, doi:10.3390/su14094966_

Round 1

Reviewer 1 Report

Although COVID-19 is a group-forming criterion in the experience of anxiety based on the title, this was unfortunately not justified by the results that highlighted in the abstract. The abstract needs to be rewritten anyway, its structural fragmentation needs to be eliminated, and more results and its practical significance can be stated here. These aspects are now very weak in the abstract.

Despite the authors point out that anxiety has a detrimental effect on performance, they do little to address the minimum level of anxiety expected before a tennis competition, making performance at its maximum. It is also an interesting question whether and how the constant of this expected value, which is probably greater than 0, was affected by COVID-19.

In any case, I am skeptical about the research issues. I don’t think the question is how age and gender have an effect on anxiety, but rather how COVID has had an effect on anxiety that can also be measured on the tennis court in different groups of players.

The expectations and statements about the role of gender in the hypotheses (boys have higher levels of self-confidence and have better stress tolerance) are random-like because these were not supported by the literature analysis and its explanation.

The results are very poorly explained, some tables are without any comments, this also needs to be supplemented. It would be good to display Figure 1 in a larger size.

It is necessary to completely rewrite the section of Conclusions as it doesn’t contain any real conclusions at all. What are the conclusions exactly in the study? Why was it important to do this research? So what are the responsibilities of coaches, organizers, competitors, etc. now?

Author Response

Response to Reviewer 1

Although COVID-19 is a group-forming criterion in the experience of anxiety based on the title, this was unfortunately not justified by the results that highlighted in the abstract. The abstract needs to be rewritten anyway, its structural fragmentation needs to be eliminated, and more results and its practical significance can be stated here. These aspects are now very weak in the abstract.

Thank you for your comment. Following your suggestion, the abstract has been modified. In addition, we agree with your position regarding COVID-19, so we have not included this aspect in the title.

Despite the authors point out that anxiety has a detrimental effect on performance, they do little to address the minimum level of anxiety expected before a tennis competition, making performance at its maximum. It is also an interesting question whether and how the constant of this expected value, which is probably greater than 0, was affected by COVID-19.

We see your interesting point. Unfortunately, performance was not included in the analysis since we focused on pre- and post-competitive anxiety. We agree that this study would be useful, so we have included it into the limitation section (in the discussion). 

In any case, I am skeptical about the research issues. I don’t think the question is how age and gender have an effect on anxiety, but rather how COVID has had an effect on anxiety that can also be measured on the tennis court in different groups of players.

After a thoughtful discussion, we have considered to remove the hypothesis regarding COVID-19. We believe that this variable has not been controlled since pre COVID assessments were not conducted (we only assessed anxiety after COVID-19 outbreak). Thus, conclusions about this topic is so limited. Instead, we have acknowledged, in both introduction and discussion sections, that anxiety values could have been influenced by COVID-19 as previous studies have reported. Furthermore, we have compared our results with a previous study which study anxiety prior to COVID-19 pandemic. Although the mean of anxiety is quite similar to the obtained in the present study, we cannot affirm that this could the same in our tennis players.

We really believe that after excluding this hypothesis from our manuscript, the results are clearer and more precise than before.

The expectations and statements about the role of gender in the hypotheses (boys have higher levels of self-confidence and have better stress tolerance) are random-like because these were not supported by the literature analysis and its explanation.

Thank you for your suggestion. We have rewritten the paragraph of the introduction regarding this topic in order to clarify why we hypothesized.

“Previous studies have studied the differences in anxiety between male and female tennis player during and precompetition. Regarding precompetitive anxiety, previous studies have reported that female athletes reported higher levels of anxiety than their male counterparts due to an increase of somatic symptoms and a decline in self-confidence before competition [50]. Similarly, Khot & Bujurke [27] found that female players had higher anxiety and stress levels than male. Cohen-Zada et al. [24] assessed the performance of female and male professional tennis players when competing in matches under pressure with high monetary rewards. During the match, men were consistently found choking under competitive pressure. However, women showed a reduction in performance in key moments, it was never 50% smaller than in the case of men. De Paola & Scoppa [25] showed that women who lost the first set would play poorly the second set much more likely than men did. Women also showed more disappointment when under pressure of receiving negative feedback and being behind. However, there are studies which did not find differences between male and female tennis players. In this line, Keskin et al. [26] did not find significant differences between adults male and female tennis players in the state anxiety (somatic and cognitive). Similarly, a previous study found that females showed less competitive anxiety than males with increasing age [16].”

The results are very poorly explained, some tables are without any comments, this also needs to be supplemented.

We totally agree that results´ description could be improved. In this regard, we have included Table 1 in this section. Furthermore, results summarized in this table have been included (which were missing in the first version). Furthermore, description of Table 5 has been improved.

It would be good to display Figure 1 in a larger size.

Thank you for your recommendation. It has been increased.

It is necessary to completely rewrite the section of Conclusions as it doesn’t contain any real conclusions at all. What are the conclusions exactly in the study? Why was it important to do this research? So what are the responsibilities of coaches, organizers, competitors, etc. now?

Thank you for your recommendation. We have rewritten the conclusion in order to be more precise.

“Players under 14 years old showed lower values of pre-competitive state anxiety and pre-competitive somatic anxiety and higher post-competitive self-confidence than players over 14 years old. Regarding gender, girls reported higher values of pre-competitive state anxiety and pre-competitive somatic anxiety than boys. Therefore, coaches and sport psychologists should implement adequate on- and off-court individualized interventions to manage anxiety, specifically in girls and players over 14 years old. Although anxiety levels were similar to those before the COVID-19 pandemic, due to the influence of the pandemic on mental health, results might be taken with caution.”

Author Response

Response to Reviewer 2

I have carefully reviewed manuscript of Martínez-Gallego et al. titled: “Gender and age influence in pre-competitive and post-competitive anxiety in young tennis players after COVID-19 pandemic".

This manuscript reports on the cross-sectional association about the influence of age and gender on pre-competitive and post-competitive anxiety and self-confidence in young tennis players and the influence of the COVID-19 pandemic. This topic is relevant. However, I have a number of concerns and suggestions that the authors should address.

Thank you for all your valuables and constructive comments. We truly believe that after considering all your suggestions, the quality of the manuscript has been significantly improved.

Introduction

The introduction is too long, also several sentences that are way too long and difficult to read. An example is in lines 45 to 52. Similar sentences appear elsewhere. I strongly suggest that the author try to shorten the introduction and long sentences so that the manuscript is readable for the audience (recommend the introduction is no more than one page).

Thank you for your suggestions. We have rewritten the introduction, in order to introduce the topic clearly and concisely. Furthermore, long sentences have been rewritten.

Results

Please try to form the table consistently. For example, the title of p-value and effect size in tables 4 and 5 can be in one row.

You were absolutely right. Following your suggestion, tables have been modified.

Materials and methods

Line 211: with values greater than or equal to.70.  Please add space between “to” and “.70”

Done.

Line 215: suggest that a value of confidence greater than .70 is acceptable. Please check if there is extra space between “that” and “a”.

Yes, it has been corrected.

Disscusion

Line 272-277:

You mentioned that “The first hypothesis, which stated that anxiety levels in young tennis players would be higher than before the COVID-19 pandemic, was not supported.” But the next sentence “Previous studies with tennis players of similar ages that analysed post-competitive anxiety [19], found very similar cognitive and somatic anxiety values to those obtained in this study.” seems like you are talking another thing (pre and post-competitive). If so, this sentence does not support your result as your result is talking about pre and under COVID-19 pandemic. Please consider it.

Thank you for your suggestion. Following the appropriate recommendation of the other reviewer, hypothesis have been reformulated regarding COVID-19. In this regard, this statement has been also modified in the discussion.

Line 302: “in which it was stated that 14 & under players”

It is not appropriate to use “&” in the scientific paper. please consider using “and”.

Corrected.

Conclusion

Line 344-345: The first sentence “This article analysed the levels of pre- and post-competition anxiety and self-confidence of young tennis players depending on age and gender.”

You mentioned before that your study aimed to investigate the influence of age and gender on pre-competitive and post-competitive anxiety and self-confidence in young tennis players and the influence of the COVID-19 pandemic. Please be consistent with the purpose of the study throughout the text (you ignore the influence of the COVID-19 pandemic).

Following your suggestion, the conclusion section has been modified as follows:

“Players under 14 years old showed lower values of pre-competitive state anxiety and pre-competitive somatic anxiety and higher post-competitive self-confidence than players over 14 years old. Regarding gender, girls reported higher values of pre-competitive state anxiety and pre-competitive somatic anxiety than boys. There-fore, coaches and sport psychologists should implement adequate on- and off-court in-dividualized interventions to manage anxiety, specifically in girls and players over 14 years old. Although anxiety levels were similar to those before the COVID-19 pan-demic, due to the influence of COVID-19 pandemic on mental health, results might be taken with caution.”

Round 2

Reviewer 1 Report

Thank you for your revision.

This manuscript is a resubmission of an earlier submission. The following is a list of the peer review reports and author responses from that submission.